# CircRNAs and miRNAs: Key Player Duo in Breast Cancer Dynamics and Biomarkers for Breast Cancer Early Detection and Prevention

**DOI:** 10.3390/ijms252313056

**Published:** 2024-12-04

**Authors:** Nour Maatouk, Abdallah Kurdi, Sarah Marei, Rihab Nasr, Rabih Talhouk

**Affiliations:** 1Department of Biology, Faculty of Arts and Sciences, American University of Beirut, Beirut 11-0236, Lebanon; nam43@mail.aub.edu (N.M.); shm32@mail.aub.edu (S.M.); 2Department of Biochemistry and Molecular Genetics, Faculty of Medicine, American University of Beirut, Beirut 11-0236, Lebanon; ak161@aub.edu.lb; 3Department of Anatomy, Cell Biology and Physiological Sciences, Faculty of Medicine, American University of Beirut, Beirut 11-0236, Lebanon; rn03@aub.edu.lb

**Keywords:** breast cancer, epigenetics, non-coding RNAs

## Abstract

Breast cancer (BC) remains a significant global health issue, necessitating advanced molecular approaches for early detection and prevention. This review delves into the roles of microRNAs (miRNAs) and circular RNAs (circRNAs) in BC, highlighting their potential as non-invasive biomarkers. Utilizing in silico tools and databases, we propose a novel methodology to establish mRNA/circRNA/miRNA axes possibly indicative of early detection and possible prevention. We propose that during early tumor initiation, some changes in oncogene or tumor suppressor gene expression (mRNA) are mirrored by alterations in corresponding circRNAs and reciprocal changes in sponged miRNAs affecting tumorigenesis pathways. We used two Gene Expression Omnibus (GEO) datasets and identified five mRNA/circRNA/miRNA axes as early possible tumor initiation biomarkers. We further validated the proposed axes through a Kaplan–Meier (KM) plot and enrichment analysis of miRNA expression using patient data. Evaluating coupled differential expression of circRNAs and miRNAs in body fluids or exosomes provides greater confidence than assessing either, with more axes providing even greater confidence. The proposed methodology not only improves early BC detection reliability but also has applications for other cancers, enhancing preventive measures.

## 1. Introduction

Cancer is a dynamic, highly heterogeneous disease [1]. Despite the dramatic improvement in cancer research, our understanding of the tumor initiation mechanisms lags far behind. The conversion from a nonmalignant to a malignant cell is now better understood due to advances in molecular diagnostics and sequencing technologies [2]. Commonly, a tumor arises through the acquisition of mutations in susceptible cell types [3]. The progression of the disease occurs specifically through the accumulation of sequential alterations which, combined, enhance genome instability that contributes to the acquisition of cancer hallmarks [4]. While genetic mutations are the primary drivers of tumor initiation and progression, a growing body of evidence implicates the fundamental role of epigenetic regulation [5].

Breast cancer (BC) is the second leading cause of cancer-related death among women [6]. BC is classified through histology, immunopathology, or mRNA and microRNA (miRNA) expression profiles. The non-invasive form of BC is ductal carcinoma in situ (DCIS). Invasive ductal carcinoma (IDC) and invasive lobular carcinoma are common invasive types that infiltrate the basement membrane (BM). BC is also categorized by receptor status: estrogen (ER), progesterone (PR), and HER2, which affects prognosis, treatment, and clinical decisions. ER-positive, HER2-negative tumors have better outcomes with hormone therapy. HER2-positive and triple-positive cancers respond well to HER2-targeted therapies; however, triple-negative cancers lack effective targeted treatments. Profiling mRNA expression of cancer cells has revealed five distinct molecular subtypes of BC: luminal A, normal-like, luminal B, HER2-enriched, and basal. Additionally, miRNA signatures have further enhanced these classifications, introducing additional subtypes that strengthen our understanding of the molecular complexity [7,8]. The improvements in screening, diagnostic techniques, and therapeutic methods have resulted in better clinical outcomes for BC [9]. Nevertheless, despite these advancements, prevention stands as the most effective tool for cancer control. However, early detection remains a challenge for earlier containment and prevention of the disease.

In this review, we provide an overview of the dysregulated expression profiles of specific miRNAs and circular RNAs (circRNAs) in BC, highlighting their contributions to oncogenesis. We discuss the roles of specific miRNAs as either oncogenes or tumor suppressors and the aberrant expression patterns of their circRNAs that disrupt normal miRNA regulatory networks, significantly influencing tumorigenic pathways. Using in silico tools and databases, we further propose a novel epigenetic-based methodology to identify mRNA/circRNA/miRNA expression axes to be used as non-invasive biomarkers for early BC detection and hence possible prevention.

## 2. Exploring Non-Coding RNAs: Understanding miRNAs and circRNAs—Their Definitions, Characteristics, and Regulatory Mechanisms

### 2.1. Definition, Characteristics, and Regulation of miRNAs

Over the last decade, miRNAs were identified as relevant regulators of various cellular functions such as metabolism, differentiation, proliferation, cell death, and migration in response to internal physiological as well as external environmental factors [10]. Approximately 22 nucleotides long, miRNAs are evolutionarily conserved, single-stranded, non-coding RNAs that modulate mRNA expression through post-transcriptional regulation [11,12]. Since their discovery in Caenorhabditis elegans, miRNAs have been found to constitute 1–5% of the human genome and regulate at least 30% of protein-coding genes [13,14]. Recent reports suggest miRNAs can also target regions within protein-coding sequences (CDSs) [15]. In humans and many other organisms, miRNA genes are located either in intergenic, intronic, or intragenic coding RNA regions within an exon of a gene. At present, almost half of the identified miRNAs are processed mainly from introns, with a minor portion being processed from exons of protein-coding genes. Intergenic miRNAs carry their own promotor and are thus independently transcribed [16,17]. In some cases, miRNAs, pertaining to the same family, are transcribed in one long transcript known as clusters, with a similar seed region [18]. The mature guide miRNA strand residing in the miRISC (miRNA-induced silencing complex) interacts with the 3′ UTR of the target mRNA [19,20]. One miRNA can have up to 200 downstream gene targets [13]. miRNA interacts with the target mRNA through sequence-complementarity-based interactions. The complementary sequence of miRNA binds to mRNA response elements (MREs) on the target mRNA [21]. The extent of MRE complementarity influences the gene silencing mechanism, whether it is an AGO2-dependent slicing of target mRNA or miRISC-mediated translational inhibition and target mRNA decay [22]. Whenever the miRNA-MRE interaction is fully complementary, mRNA cleavage is induced through the activation of AGO2 endonuclease [22]. This interaction destabilizes the AGO association with the 3′ end of the miRNA, initiating its degradation [23,24]. Commonly, the miRNA-MRE interaction is not fully complementary in animal cells [25]. AGO2 endonuclease activity is restrained through at least one central mismatch in the MRE-miRNA complex, acting as an RNA interference mediator. miRNA-MRE interaction takes place in the 5′ seed region. The interaction is stabilized via an additional pairing at the 3′ end which aids in the stability and specificity of the miRNA-MRE interaction [26]. Many factors regulate the miRNA-mediated gene regulation; among these are compartmentalization and functionality of miRISC, the abundance and availability of miRNAs, and tissue/cell-specific alternative splicing mechanisms affecting 3′ UTRs [27,28,29].

### 2.2. Definition, Characteristics, and Regulation of circRNAs

The expression of target mRNA can also be regulated through competing endogenous RNAs (ceRNAs), such as with circRNAs [30] which act by sequestering miRISC from target mRNA. CircRNAs, produced through head-to-tail back splicing, are single-stranded RNA molecules with a covalently closed loop structure. Unlike their linear mRNA counterparts, they lack a 5′ cap and a 3′ poly(A) tail, which provides them with stability and a longer lifespan due to resistance against RNA exonucleases [31]. CircRNAs are conserved across species ranging from viruses to mammals [32]. They arise from a variety of genomic sources and range from exonic circRNAs, intronic circRNAs, exon–intron circRNAs, and intergenic circRNAs. Exonic circRNAs majorly act as potential regulatory candidates or peptides coding RNAs. Intronic circRNAs are formed from intronic sequences of a gene, and once an intron is retained during splicing, it forms a circular structure. Intronic circRNAs essentially regulate gene expression by sequestering splice factors or other RNA-binding proteins. Exon–intron circRNAs are formed through the back splicing of exonic sequences with intronic sequences. These circRNAs can consequently act as gene regulators and protein-coding RNAs. Finally, intergenic circRNAs are formed from sequences located between genes. Back-splicing events involving exons from different genes result in the circularization of intervening sequences. Studies have addressed a wide range of circRNA cellular functions ranging from gene transcription and expression regulation and protein scaffolds to miRNA binding/sponging [33,34,35,36]. Interestingly, recent studies are shedding light on identifying translatable circRNAs [37]. Indeed, some circRNAs can be translated into functional polypeptides; however, the existing research on circRNA polypeptides is quite limited. As previously mentioned, circRNAs act as ceRNAs or miRNA sponges. circRNAs contain specific complementary sequences to miRNAs known as miRNA response elements (MREs) or miRNA binding sites like those described on mRNA targets of miRNA [38]. Once recognized and bound, miRNA becomes “trapped” or “sponged” and thus sequestered within the circRNA-miRNA complex which in turn prevents the miRNA from binding to its downstream mRNA target. Sponged miRNAs can no longer interact with their target mRNAs, which can lead to an increase in the expression of target genes through “derepressing” them [39]. Given the variety of the length of circRNAs, the number of potential MREs they contain varies. This variability is linked not only to the length of the circRNAs but also to their specific nucleotide sequences. Multiple circRNAs can contain similar or identical MREs for a specific miRNA. Also, the same miRNA may be sponged by different circRNAs. miRNA interaction is influenced by several factors such as sequence complementarity, the number and affinity of MREs, circRNA and miRNA abundance, and cellular context. Notably, circRNAs and miRNAs are found within extracellular vesicles (EVs), which play a critical role in facilitating their stability and transport between cells.

### 2.3. CircRNAs and miRNAs in EVs

EVs are small membrane-bound particles released by cells into the interstitial fluid. They are rich sources of molecules such as proteins, lipids, DNA, RNA, and ncRNAs, providing these molecules a protective environment that extends their stability in body fluids. EVs play key roles in intercellular communication through the transportation of their content to recipient cells, affecting various physiological processes. EVs are classified into several subtypes, including exosomes, microvesicles, and apoptotic bodies, each varying in biogenesis and composition [40]. EVs are gradually more recognized for their potential as biomarkers for disease diagnosis and prognosis, specifically cancer, as well as their therapeutic applications. Exosomes, or small EVs (sEVs), are the most extensively studied EVs. Exosomes are known to regulate the sensitivity of cancer cells to chemotherapeutic agents by transporting non-coding RNAs such as miRNAs, long non-coding RNAs (lncRNAs), and circRNAs [40]. Thus, miRNAs and circRNAs contained within exosomes are released to the interstitial space and eventually to a variety of body fluids including blood, plasma and serum, saliva, urine, breast milk, and cerebrospinal fluid [41,42]. Circulating miRNAs and circRNAs, whether free or encapsulated within exosomes and other EVs, are considered potential promising circulating biomarkers in liquid biopsies [43,44]. The altered expression levels of free or encapsulated miRNAs and circRNAs could reflect disease initiation and progression in a non-invasive manner. Further investigation is required to unravel the potential of EV-derived non-coding RNAs in predicting disease states and boosting the precision and efficacy of liquid biopsy.

## 3. Exploring circRNA/miRNA Dynamics in BC

### 3.1. Alteration of miRNA Expression in Cancer

miRNA expression is under the control of a variety of mechanisms, and alterations in these regulatory processes can lead to a range of human diseases, including cancer [45]. The discovery of miRNA genes miR-15a and 16-1 in a chromosomal region commonly deleted in chronic lymphocytic leukemia (CLL) indicates that miRNAs can be in genome regions prone to cancer-related alterations [46,47,48,49]. Inherited mutations in the primary transcripts of miR-15a and miR-16-1 and single nucleotide polymorphisms (SNPs) have been reported to cause reduced expression of these miRNAs in CLL [50,51]. Apart from genetic changes, defects in the miRNA biogenesis machinery can also modulate miRNA expression. For example, adenosine deaminase acting on RNA (ADARs) can influence the expression of miRNAs that recognize adenosine residues. In the case of miR-142, the editing of pri-miR-142 by ADARs leads to a reduction in its processing by Drosha, followed by degradation within the RISC complex, ultimately resulting in lower levels of the mature miR-142 [52]. Variations in miRNA levels due to altered activity of Drosha or Dicer have been observed across various types of tumors. Specifically, the silencing of either Dicer or Drosha is linked to increased cellular transformation and tumorigenesis. For example, the downregulation of Dicer expression in BC cell lines showed a correlation with the metastatic spread of the tumor [53]. Furthermore, the processing of miRNAs can be influenced by other miRNAs in a process known as “complex reciprocal interaction and regulation”. Tang et al. reported that miR-709 is found in the nucleus of various cell types in mice, where it binds to pri-miR-15a/16-1. The miR-709-pri-miR-15a/16-1 interaction prevents the processing of pri-miR-15a/16-1 into its precursor form, pre-miR-15a/16-1, thus blocking its maturation [54]. In BC, the interaction between miR-709 and p53 pathway components significantly affects tumor growth and resistance to treatment [55,56]. The altered expression of miRNA in cancer can also be linked to epigenetic modifications, such as changes in DNA methylation. Studies of miRNA genes show that half of these are associated with CpG islands [57]. Research by Saito et al. found that changes in methylation status can lead to the silencing of tumor suppressor miRNAs. This was shown by the significant upregulation of miR-127 in bladder cancer cell lines and human fibroblasts treated with the *DNMT* inhibitor 5-Aza-2′-deoxycytidine. Typically silenced in various cancers, miR-127 targets the proto-oncogene BCL-6 and features a CpG island promoter [58]. Similarly, miR-9-1 was shown to be downregulated subsequent to hypermethylation in BC [59]. Poli et al. showed higher methylation levels in basal BC cells compared to luminal BC cells which negatively correlates with miR-29c expression levels and shows that methylation-derived miRNA alternation in expression is subtype-specific [60,61]. Methylation is not the only epigenetic mechanism impacting miRNA expression; histone deacetylase (HDAC) inhibition also plays a significant role [62]. The relationship between miRNAs and epigenetics is quite complicated since miRNAs themselves can regulate components of the epigenetic machinery, creating a complex feedback loop [63,64].

The alteration of miRNA expression in cancer is a complex process influenced by various genetic, epigenetic, and biogenesis-related factors. Depending on whether oncogenic miRNAs (known as “oncomiRs”) or tumor suppressor miRNAs (known as “oncosuppressor miRs”) are involved, these alterations can lead to either the promotion or suppression of tumor growth and progression. The study of miRNAs in vitro and their implications in population studies provides a more comprehensive understanding of their role in cancer progression and potential therapeutic applications.

### 3.2. miRNA In Vitro and Population Studies in BC

Table 1 summarizes miRNAs in BC studies, categorized as tumor suppressors or oncomiRs in both in vitro and in vivo models, and highlights their regulation patterns, study models, and target genes. miR-1 was shown to act as a tumor suppressor miRNA in human MCF-7 BC cells as well as MDA-MB-435, MDA-MB-468, and MDA-MB-231 cells. miR-1 inhibits tumor growth and metastasis of BC cells through the downregulation of the *CDK4* gene involved in the cellular transition from the G1 phase to the S phase. The outcome of such inhibition in BC cells results in blocked cellular proliferation. *TMSB4X*, *CNN3*, *TWF1*, *CORO1C*, and *WASF2* are also downstream genes of miR-1. These genes are potential regulators of actin filaments and thus promote metastasis in malignant tumors. The loss of *TMSB4X*, *CNN3*, *TWF1*, *CORO1C*, and *WASF2* genes through miR-1-mediated downregulation inhibits cellular migration and invasion in BC cell lines [65]. Wang et al. showed that the transfection of BC cell lines T47D and SKBR3 with mimics of miR-125-b, another tumor suppressor miRNA, results in reduced *MMP11* protein expression. Downregulation of *MMP11* stimulates cancer cell apoptosis and limits tumor cell invasion and metastasis [66]. In addition to metastasis and cellular invasion, angiogenesis is another cancerous mechanism reported to be under the regulation of miRNA-mediated gene silencing. miR-100, enriched in MSC-derived exosomes, showed an inverse correlation with the vascular endothelial growth factor (*VEGF*) expression in MDA-MB-231 and MCF-7 cells through modulating the mTOR/HIF-1α signaling axis [67]. Interestingly, VEGF was also inversely correlated with miR-126 in MCF-7 BC cells. miR-126 expression has been linked to reduced cellular proliferation and cellular arrest at the G1 phase [68]. miR-355 and miR-145 were reported to be significantly downregulated in BC cell lines compared to controls. The use of miR-145 mimics significantly reduced the level of endogenous *ROCK1* mRNA and proteins in transfected cells compared to non-transfected cells. *ROCK1*, a member of the Rho family, is a crucial serine/threonine kinase involved in controlling the arrangement of the actin cytoskeleton. Increased *ROCK1* expression has been observed in various BC cell lines, which in turn correlates with cellular movement and invasion [69].

The preceding studies shed light on various tumor suppressor miRNAs differentially expressed in vitro in BC models. Nevertheless, oncomiRs contribute to tumor initiation and progression. Among these, miR-21 is reported to act as an oncomiR in human BC cell lines including Hs578T, MDA-MB-231, SK-BR-3, and MCF-7. A study by Wang et al. showed that blocking miR-21 suppresses cellular proliferation and metastasis in BC cell lines [70]. The study reports a new target gene for miR-21 which is the Leucine zipper transcription factor-like 1 (*LZTFL1*), a key gene regulating cancer metastasis [71]. LZTFL1 acts as a tumor suppressor gene by regulating the β-catenin signaling pathway, and once lost, it activates the Epithelial-to-Mesenchymal Transition (EMT)-related pathways in several types of cancer [71,72]. Within the group of oncomiRs, miR-155 plays a crucial role in downregulating the mRNA target of the Tetraspanin-5 (*TSPAN5*) gene in the MCF-7 BC cell line. miR-155 regulates cellular proliferation, apoptosis, and autophagy. Inhibition of miR-155, using an antisense sequence delivered in gold nanoparticles, led to significant restoration of the normal cellular function of the miRNA downstream target gene [73]. Additionally, a study by Guo et al. found that MCF10A cells that overexpressed miR-10b displayed enhanced motility and proliferation [74].

miRNA expression is stage- and tissue-specific [82]. Interestingly, the same miRNA can act as a tumor suppressor miRNA in one tissue and an oncomiR in another [83]. miR-17, for example, is reported to act as an oncomiR in B-cell lymphoma and as a tumor suppressor miRNA and oncomiR in BC cells [84,85]. Even in the same cancer, miR-17-92 is context- and subtype-dependent. miR-17-92 expression is upregulated in triple-negative breast cancer (TNBC) but downregulated in estrogen receptor (ER)-positive BC [86]. In vitro models, while useful for initial screening and understanding basic mechanisms, often fail to capture complex cellular interactions and mimic the actual microenvironment of tumors. Consequently, these models may not fully represent the dynamics of miRNA regulation in the actual breast tissue.

In vivo studies better integrate the various microenvironmental factors involved in miRNA regulation and pave the way for developing more effective therapeutic strategies in population-targeted cancer research. A study by Gong identified 102 estrogen receptor (ER)-subtype-related differentially expressed miRNAs in breast tumors, predominantly race-specific, with only 23 differentially expressed miRNAs common between African Americans (AAs) and European Americans (EAs). This suggests unique miRNA profiles in ER-negative and ER-positive tumors among different ancestries, reflecting variations in tumor biology and BC heterogeneity [75]. A study by Shishavan et al. on an Iranian population described the link between miR-125a-5p and STAT3 levels in BC patients’ tissues. The study found that miR-125a-5p expression is downregulated in BC tissues compared to normal tissues. On the other hand, *STAT3* expression levels were higher in BC tissues compared to normal tissues, suggesting an inhibitory effect of miR-125a-5p on *STAT3* [76]. Another population study by Nassar et al. examines the miRNA microarray profiles in BC tissues compared to normal adjacent tissues obtained from Lebanese patients diagnosed with early-stage BC [77]. In total, 173 miRNAs were found significantly dysregulated, and 74 miRNAs exhibited a fold change greater than 2, with 17 of them being reported for the first time in BC. Another study by Tfaily et al. showed an alteration in the expression of five miRNAs in North African and Algerian BC tissue samples compared to normal adjacent tissue [78]. miR-183, miR-182, miR-21, miR-200c, and miR-425-5p were found to be significantly upregulated in Algerian BC tissues compared to normal tissues. The reported miRNAs were also previously found to be dysregulated in other populations, among which are the American, Australian, Chinese, Italian, Lebanese, Spanish, and Taiwanese populations [77,87,88,89,90,91,92,93].

Circulating miRNAs in biological fluids are stable and easily detectable by PCR-based approaches, rendering them effective biomarkers for BC. Diansyah et al. reported the overexpression of miR-21 in the plasma of 26 adult Indonesian female early-stage BC patients (stage 1A, 1B, 2A, and 2B) compared to 16 healthy samples, suggesting that circulating miR-21 serves as a potential predictive tool for the detection of early BC initiation events [79]. Similarly, a comprehensive study by Wu et al., encompassing 11 published clinical studies and 921 BC patients, showed that miR-1246 and miR-21 were enriched within exosomes and exhibited higher expression levels in BC patients compared to the normal condition. Additionally, exosomal miR-340-5p, miR-17-5p, miR-130a-3p, and miR-93-5p have been proposed to be associated with tumorigenesis and metastasis [80]. Further, circulating miRNA-148a and miRNA-30c were downregulated in 75 female BC patients compared to 55 healthy controls, indicating their potential as diagnostic biomarkers in BC patients [81].

### 3.3. Alteration of circRNA Expression in Cancer

miRNAs are one component of the broader epigenetic landscape; thus, their expression alone offers limited information about the overall epigenetic profile during tumorigenic events. In 1991, Nigro et al. identified circRNA transcripts from the *DCC* gene (the human Deleted Colon Cancer gene) [94]. Circular transcripts were first described in several genes, including *ETS-1* [95], *human cytochrome P450* [96], *dystrophin* [97], and antisense non-coding RNA at the *INK4* locus [98]. These transcripts were initially thought to be just splicing errors with no biological relevance. However, with advancements in RNA-seq and bioinformatics, researchers began to identify more of these circRNAs. Salzman et al. discovered many circRNA transcripts in pediatric acute lymphoblastic leukemia samples [99]. An alteration in circRNA expression correlates with tumorigenic events [100]. Altered circRNA expression could be due to changes in splicing factors involved in circRNA synthesis. One common example is the alteration of circRNA expression during the EMT in immortalized breast epithelial cells by an RNA-binding protein (RBP). QKI, an RBP, boosts the production of various circRNAs by binding to specific recognition sites within introns near the splice sites where circRNAs are formed [101]. Modulation in transcription precursors of circRNAs can change their expression levels, correlating with a wide range of cancers. Other RBPs, such as DHX9, destabilize the formation of circRNA through preventing the formation of a stable double-stranded RNA region. Being an RNA helicase, DHX9 unwinds double-stranded RNA regions required for circRNA biogenesis. Like miRNAs, circRNAs play a dual role in tumor-suppressing and tumor-promoting function. A high level of circHMCU, hsa_circ_0055478 [circPTCD3], hsa_circ_0005728 [circUBE2D2], hsa_circ_0087784 [circRNF20], hsa_circ_008717 [circABCB10], hsa_circ0005230, and circRPPH1_015 correlates with BC diagnosis and prognosis [102,103,104,105,106,107]. In parallel, other circRNAs show tumor suppressor activity. CircCCDC85A and hsa_circ_0072309 were shown to be downregulated in BC tissues as well as BC cell lines when compared to the normal condition [108].

### 3.4. CircRNA In Vitro and Population Studies in BC

Table 2 summarizes circRNAs in BC studies, categorized as tumor suppressor circRNA or oncocircRNA in both in vitro and in vivo models, and highlights their regulation patterns and study models. CircRPAP2 (hsa_circ_0000091) was shown to be downregulated in BC cell lines and tissue samples. A study showed that circRPAP2 inhibits the proliferation and migration of BC cells in vitro. CircRPAP2 binds to SRSF1, an oncoprotein, which alters the SRSF1-*PTK2* pre-mRNA interaction. This inhibition attenuates SRSF1’s alternative splicing of PTK2, which is essential for its oncogenic activity, leading to the downregulation of *PTK2* mRNA and protein levels [109]. Another study reported that circ-1073 expression is downregulated in BC cells (BCCs) and tissues compared to normal mammary epithelial cells. Low levels of circ-1073 were associated with poor BC patients’ prognosis. When circ-1073 is highly expressed, it inhibits proliferation in various BCCs and induces apoptosis by increasing the cleavage of Caspase-3/9, while reducing cell mobility and EMT. Intratumoral injection of nanoparticles carrying a circ-1073 plasmid resulted in the inhibition of xenograft tumor growth [110]. Another study found that circ-VRK1 expression is lower in BC cell lines like BT474, MDA-MB-453, and MDA-MB-231 compared to the normal breast epithelial cell line MCF10A. Circ-VRK1 limited cellular proliferation and promoted apoptosis in the MDA-MB-231 cell line. Circ-VRK1 was shown to be downregulated in 350 BC tissues compared to 163 adjacent normal tissues. Circ-VRK1 expression was associated with smaller tumor size, reduced T stage, and lower Tumor, Node, Metastasis (TNM) stage [111].

To investigate the mechanisms underlying the role of circRNA in TNBC, another study established stable cell lines of MDA-MB-231 and BT-549 TNBC cells expressing circ-ITCH via lentiviral vectors. The overexpression of circ-ITCH via lentiviral vectors was found to markedly inhibit TNBC proliferation, invasion, and metastasis in both cell cultures and animal models. Mechanistically, circ-ITCH was identified as a sponge for miR-214 and miR-17, leading to the increased expression of its linear isoform, ITCH. This, in turn, resulted in the inactivation of the Wnt/β-catenin signaling pathway [112]. On the other hand, Chen et al. revealed that circWSB1 expression was significantly elevated in BC MCF-7 cells. High expression of circWSB1 facilitated BC cell proliferation in BC MCF-7 cells. Hypoxia-induced HIF1α expression results in an upregulation of circWSB1 expression [113]. On the other hand, a study found that circ-Dnmt1 expression was upregulated in eight BC cell lines as well as in breast carcinoma patients. Silencing circ-Dnmt1 in MDA-MB-231 and BT-549 cells inhibited cellular proliferation and survival, while high circ-Dnmt1 expression enhanced BC cell proliferation and survival by promoting cellular autophagy [114]. Circ-Dnmt1-mediated autophagy is crucial for the prevention of cellular senescence and tumor growth. Further, high circ-Dnmt1 expression levels interact with both *p53* and *AUF1*, stimulating their translocation into the nucleus. In the nucleus, p53 induces autophagy, while AUF1 stabilizes *Dnmt1* mRNA, increasing its translation. Active Dnmt1 then enters the nucleus and suppresses p53 transcription. In addition, an interesting in vivo study found that the expression of hsa_circ_0001831 led to low SCRIB mRNA expression levels but increased BC cell proliferation, migration, and invasion in mice. However, the exonic sequence did not affect *SCRIB* mRNA splicing but decreased its translation into protein, resulting in higher E-cadherin levels and lower N-cadherin and Vimentin levels, which enhanced cellular migration, invasion, proliferation, colony formation, and tumorigenesis [115].

A meta-analysis study conducted with a total of 29 studies involving 4405 patients identified differentially expressed circRNAs through searches in PubMed, Embase, Web of Science, CNKI, and Cochrane Library. The analysis revealed that high expression levels of tumor-promoter circRNAs such as circSEPT9, hsa_circ_0001785, hsa_circ_0108942, circKIF4A, and circEPSTI1 were associated with a shorter survival time, while tumor suppressor circRNAs such as circ-1073, circ-VRK1, hsa_circ_0068033, circTADA2A-E6, and circRNA_103809 were linked to a favorable prognosis. Additionally, elevated levels of oncogenic circRNAs correlated with poor clinical outcomes, whereas tumor suppressor circRNAs exhibited the opposite trend [116]. Another study has investigated the expression levels of hsa_circ_0005046 and hsa_circ_0001791 in 60 BC tissues and their matched adjacent normal tissues. The results demonstrated significantly higher expression levels of hsa_circ_0005046 and hsa_circ_0001791 in BC tissues compared to the paired adjacent normal tissues [117]. A study by Zhang et al. highlighted abnormal circRNA expression profiles in the peripheral blood of BC patients. Microarray analysis revealed 41 dysregulated circRNAs with a fold change of 2 or more, including 19 upregulated and 22 downregulated circRNAs. Among these, three circRNAs (hsa_circ_0001785, hsa_circ_0108942, and hsa_circ_0068033) were significantly dysregulated and further validated by RT-PCR. Plasma levels of hsa_circ_0001785 correlate with histological grade, TNM stage, and metastasis [118]. Chen et al. described the role of circSEPT9 in the development and progression of TNBC by examining circRNA expression patterns in four pairs of TNBC tissues compared to their corresponding normal tissues. They found that high expression of circSEPT9 in TNBC tissues correlates with advanced clinical stages and poor prognosis. Knockdown of circSEPT9 inhibited the proliferation, migration, and invasion of TNBC cells, inducing apoptosis and autophagy and suppressing tumor growth and metastasis in vivo. In contrast, the upregulation of circSEPT9 had the opposite effect [119].

In the preceding paragraphs, we explored the differentially expressed circRNAs and miRNAs associated with BC, shedding light on the intricate molecular landscape of this disease. However, in modern healthcare, early detection and prevention are essential pillars. Among the reasons for which these early detection measures are fundamental are the improved treatment outcomes, delay in disease progression, and focus on preventive measures. Thus, to gain deeper insights into the early stages of BC, it is imperative to understand the networks that exist between circRNAs and their target miRNAs which characterize the initiation of this disease.

## 4. A New Paradigm: circRNA and miRNA Interplay in Advancing Early Detection and Prevention

### 4.1. The Significance of Early Detection and Prevention

Early detection of various types of cancers is crucial for effective treatment and improved patient outcomes. Recent studies, including whole exome sequencing of different cancers, reveal a widespread presence of mutations in genes that control the epigenome [120]. Both circRNAs and miRNAs are studied as early biomarkers of cancer due to their altered expression during tumor initiation events and stability in body fluids [121,122,123]. Using total RNA and small RNA sequencing, Rao et al. reported upregulation of hsa_circ_0006743, hsa_circ_0002496, and hsa_circ_0023990 in early-stage I-IIA BC tissues compared to the normal condition. Interestingly, the study also reported an opposite correlation in expression between hsa_circ_0001946 and miR-26a, miR-342, miR-30b, miR-214, miR-125b, miR-196a, miR-92b, and miR-19a, as well as a reciprocal expression between hsa_circ_0002496 and miR-125B1, miR-204, and miR-10B. Further, the miRNAs were all found to be involved in BC development, suggesting the implication of circRNA/miRNA in early tumorigenic events [124]. The reciprocal correlation proposes a possible potential sponging and integrated signaling pathways between a circRNA and its corresponding respective potentially sponged miRNA. Due to sharing miRNA binding sites with mRNAs, and through competing with ceRNAs for binding to miRNAs, circRNAs indirectly regulate the expression of miRNAs’ downstream genes thus regulate the post-transcriptional expression of those genes. This interplay can exert a significant effect on a multitude of cellular processes intricately influencing the dynamics of BC initiation and progression. Previous studies have shed light on potential mRNA/circRNA/miRNA axes involved in tumorigenic events [125,126,127,128].

### 4.2. mRNA/circRNA/miRNA Axes

The expression of circRNAs, proposed to sponge miRNAs and block their activity, is largely dependent on the expression levels of the protein-coding genes from which they arise. This is hypothesized to result in an aligned change in levels of circRNAs-derived genes, during the pre-oncogenic stage, as well as, reciprocally, in corresponding sponged miRNAs. Indeed, two possible scenarios can take place: either the upregulation of oncogenic circRNAs which downregulate the expression of tumor-suppressive miRNAs or the downregulation of tumor-suppressive circRNAs which upregulate the expression of oncomiRNAs.

Upregulated oncogenic circRNAs were found to be, through tumor-suppressive miRNA sponging, involved in TNBC tumorigenic phenotypes such as cellular viability and cell cycle entry [105,129,130], inhibited cellular apoptosis [131,132], cellular migration and invasion in BC and specifically TNBC [133,134], energy metabolism in TNBC [135,136], stem cell activity, angiogenesis, and drug resistance [102,137]. For example, Liu et al. reported the involvement of the circ_0008039/miR-432-5p/E2F transcription factor 3 (E2F3) axis in promoting cellular proliferation in BC cell lines (BT-20, BT-474, MCF-7, BT549, MDA-MB-231, and SKBR-3) through accelerating the transition of the cell cycle from the G0/G1 to the S phase [105]. Also, the circ-ABCC4/miR-154-5p axis was found to inhibit apoptosis in human BC cell lines, MDA-MB-231 and MCF-7 [132]. The high expression of tumor-suppressive circRNAs results in increased sponging of free oncomiRs and thus limits the capacity of cancer cell proliferation and inhibits EMT [138,139]. Consequently, epithelioid cancer cells undergo a loss of polarity and tight intracellular junctions, followed by a transition to a mesenchymal phenotype, acquire stem-like molecular features, and become more resistant to drugs used for BC treatment.

The downregulation of oncosuppressor circRNAs showed an increase in the levels of free oncomiRs, which caused the activation of cancer-related pathways. Low expression of tumor suppressor circRNAs is associated with large tumor sizes, lymph node metastasis, and poor prognosis [140]. Numerous studies provided evidence that circRNAs with tumor-suppressive function induce a negative regulatory effect on the malignant behavior of BC. In TNBC, nuclear receptor subfamily 3 group C member 2 (*NR3C2*)-derived circRNA exhibits anti-metastatic properties. *NR3C2*-derived circRNA sponges miR-513a-3p, upregulating the E3 ubiquitin ligase (*HRD1*), which in turn targets Vimentin, a protein integral to EMT. *NR3C2*-derived circRNA is crucial in preventing EMT, thus reducing cellular invasion and migration [141,142,143]. Ye et al. demonstrated that the circ_0001451 (circ_FBXW7), which arises from exons 3 and 4 of the FBXW7 gene located on chromosome 4q31.3, sponges miR-197-3p. This circRNA encodes the FBXW7-185aa protein, which inhibits proliferation and migration in TNBC cells by upregulating *FBXW7*, a tumor suppressor gene, leading to the degradation of the c-Myc protein [141,142,143,144]. In addition, Huang et al. proposed the hsa_circ_0086735/miR-1296-5p/*STAT1* axis in luminal-subtype BC through microarray data analysis [125]. We have previously described the involvement of the gap junction in post-transcriptional axes and revealed Cx43/hsa_circ_0077755/miR-182 as a potential biomarker signature axis in a breast epithelial 3D culture model for a heightened risk of BC initiation [145]. Accordingly, loss of Cx43 expression in breast epithelium is associated with cell multilayering and polarity disruption, as well as a loss of the expression of hsa_circ_0077755, one of the Cx43-derived circRNAs. Consequently, the expression of miR-182, proposed to be sponged by hsa_circ_0077755, was elevated, which led to the upregulation or downregulation of the genes involved in the signaling cascade of miR-182-regulated tumorigenesis pathways [145]. When considering in vivo models, the redistribution of Cx43 and loss of epithelial polarity are observed in obese women, which is correlated with a heightened risk of BC development. CircRNA/miRNA interaction axes hold potential as targets for the prevention, diagnosis, and treatment of BC. CircRNA/miRNA axes serve as potential stage- and tissue-specific liquid biomarkers for tracking tumor dynamics [44,146].

In the section below, we use bioinformatics resources in the context of BC research. By integrating databases and in silico tools, we propose a methodology relying on the interaction of circRNAs and miRNAs to be utilized for identifying axes useful for early risk prediction, detection of BC, and preventative options.

## 5. Integration of Databases and In Silico Tools for circRNA and miRNA Data Collection and Studying circRNA/miRNA Axes in BC: A Proof of Concept

We propose that, hypothetically, the expression of circRNAs, which is in sync with that of the parent genes, inversely, though not universally, correlates with the expression of miRNAs they sponge. The screening for inverse reciprocal expression patterns of oncosuppressor or tumor suppressor circRNAs and miRNAs combined could provide valuable biomarker tools for early detection, and possibly prevention. Even in the absence of direct sponging, identifying and validating such reciprocal expression levels could provide critical insights into the molecular mechanisms regulating these candidates in tumor initiation phenotypes. The level of confidence of screening is further amplified when integrating multiple circRNA/miRNA axes. This enhances reliability and specificity for identifying early tumorigenic events. Such an approach offers a multi-level perspective on the molecular pathogenesis of cancer.

To build the circRNA/miRNA axes (Figure 1), we first used GEO (accession number: GSE182471), specifically a dataset of five BC patients and five normal breast tissues (Table 3; circRNA), to obtain the differentially expressed circRNAs and the genes from which they arise. The differential expression of the initial genes in cancer tissues was validated using UALCAN [147,148], a TCGA-based online tool. We selected the top differentially expressed circRNAs. Then, using CircInteractome [149], we predicted the list of miRNAs sponged by each differentially expressed circRNA. Only miRNAs with high context percentile (high predictive circRNA-miRNA binding potential based on sequence complementarity) are considered. The expression of these miRNAs is hypothesized to be reciprocal to that of circRNAs. This hypothesis can be confirmed through two steps using available in silico tools. The first method involves analyzing miRNA-microarray data from GEO (accession number: GSE154255) of 10 BC tissues compared to adjacent normal tissues (Table 3; miRNA). The second method involves using Kaplan Meier (KM) Plotter [150], which predicts the survival of BC patients, to examine whether the lower survival analysis of the corresponding miR gene in breast tumor patients in tissue correlates with previously predicted miRNA expression. The predicted expression must be correlated with lower survival rates among BC patients in KM Plotter. Additionally, miRNA target genes can be obtained using the “multimiR” [151] package in R-studio. Webgestalt was then used to select the miRNAs enriched in cancer-related pathways (Table 4).

Five circRNA/miRNA axes were constructed as proof of concept (Table 5). Looking into the top DE circRNAs in BC tissues compared to the normal condition, the following circRNAs were downregulated: hsa_circ_0002599 and hsa_circ_0088251. On the other hand, the following circRNAs were upregulated: hsa_circ_0044556, hsa_circ_0001875, and hsa_circ_0001414.

The five corresponding parental genes from which the aforementioned circRNAs are derived are (1) *FUK* (L-fucose coding gene, mediating cell–cell interactions), (2) *PAPPA* (this gene encodes a secreted metalloproteinase which cleaves insulin-like growth factor binding proteins (IGFBPs) and is involved in cellular inflammation), (3) *COL1A1* (Type I collagen is a member of group I collagen (fibrillar forming collagen)), (4) *FAM120A* (this gene encodes a protein that is involved in mRNA transport within the cytoplasm and plays a crucial role in oxidative-stress-induced survival signaling by activating Src family kinases and facilitating the phosphorylation and activation of PI3-kinase), and (5) *TMEM165* (this gene encodes a transmembrane protein that is crucial for maintaining cellular calcium homeostasis and glycosylation processes in the Golgi apparatus). These parental genes are hypothesized to be aligned in expression with their circRNAs. Indeed, the differential expression of the parental gene in cancer tissues which aligned with the circRNAs was validated; *FUK* and *PAPPA* showed a significant downregulation, while *COL1A1*, *FAM120A*, and *TMEM165* showed an upregulation in UALCAN primary tumor tissues compared to the normal condition (*p*-value of the differential expression of the parental genes in BC compared to the normal condition attached in Table 5). We then selected, using the circular RNA interactome, the list of miRNAs predicted to be sponged by each differentially expressed circRNA. The miRNAs were selected based on an expression profile reciprocal to that of circRNAs. Hsa-miR-326-3p, upregulated in 10 BC tissues compared to adjacent normal tissues, is shown to be sponged by the two downregulated circRNAs hsa_circ_0002599 and hsa_circ_0088251. On the other hand, hsa-miR-145-5p and hsa-miR-99a, downregulated in 10 BC tissues compared to adjacent normal tissues, are hypothesized to be sponged by the upregulated circRNAs: hsa_circ_0044556, hsa_circ_0001875, and hsa_circ_0001414. The high expression of hsa-miR-326-3p is significantly correlated with lower BC patient survival (Appendix A). Also, the low expression of hsa-miR-145-5p and hsa-miR-99a is correlated with lower BC patient survival (Appendix A). The miRNA hsa-miR-326-3p regulates a suite of downstream genes, including *SMO*, *NOTCH1*, *ERBB2*, *NF2*, *KRAS*, *ARRDC1*, *RBM47*, *CCND1*, *GYS1*, and *SP1*, among others. These genes play crucial roles in BC pathogenesis, primarily influencing the Hedgehog and NF-KB signaling pathways, as evidenced by the enrichment plots provided in the Appendix A. Similarly, hsa-miR-145-5p influences key genes such as *PARP8*, *IRS1*, *MYC*, *FSCN1*, and others, which are integral to the TGF-beta signaling pathway, angiogenesis, and various cancer-related pathways. Finally, hsa-miR-99a modulates genes such as *CCND2*, *GRB2*, *MCM4*, *LAMTOR1*, *RPS6*, and others, impacting the cell cycle, TGF-beta signaling pathway, and DNA damage response (Appendix A).

From the above, looking at the top 20 DE circRNAs, and using a combination of databases and in silico tools, we identified five novel mRNA/circRNA/miRNA axes as a proof of concept: (1) the *FUK*/hsa_circ_0002599/hsa-miR-326-3p axis, (2) the *PAPPA*/hsa_circ_0088251/hsa-miR-326-3p axis, (3) the *COL1A1*/hsa_circ_0044556/hsa-miR-145-5p axis, (4) the *FAM120A*/hsa_circ_0001875/hsa-miR-145-5p axis, and (5) the *TMEM165*/hsa_circ_0001414/hsa-miR-99a axis (Table 4). The proposed axes, and others combined, once validated, can be used as non-invasive biomarker tools in multi-dimensional screening for cancer prediction and early detection.

## 6. Evaluating mRNA/circRNA/miRNA Axes: Current Limitations, Methodological Considerations, and Future Prospects

To refine our methodological approach and strengthen the conclusions drawn, we discuss key methodological considerations and limitations related to the proposed axes, as well as future directions to enhance their reliability and applicability in cancer research.

In our analysis, we utilized different GEO datasets to identify DE circRNAs and their parental genes and miRNAs, which originate from two separate patient populations. We acknowledge the importance of consistency in the source of patient samples to reduce variability, especially given the tumor heterogeneity often observed in cancer studies [152,153,154,155]. This decision, however, was due to the current limitations in available datasets containing both circRNA and miRNA data from the same patients. To address this limitation, we followed a multi-step cross-validation process. We cross-validated the expression of the parental genes, from which the DE circRNAs arise, with patients’ data from TCGA datasets using UALCAN (with DE analysis performed between normal samples (*n* = 114) and primary tumor samples (*n* = 1097)) (Table 5 and Appendix A). We then used an independent GEO dataset to identify DE miRNAs. To link these miRNAs to our identified circRNAs, we used CircInteractome to predict miRNAs likely to be sponged by the selected circRNAs with a high context percentile. Although many of the identified circRNAs had only a single miRNA binding site, we considered factors such as the binding affinity and stability of the interaction between a circRNA and miRNA, which can enhance the biological impact of even one single binding site. The relative expression levels of the circRNA and miRNA are also critical. When the circRNA is abundant and the miRNA is present in limited quantities, even a single binding site can reduce the pool of free miRNA, impacting downstream regulatory pathways. Additionally, the targeting of each of some miRNAs, such as hsa-miR-326-3p and hsa-miR-145-5p (Table 5), by multiple circRNAs might enhance their regulatory potential, even when each is limited to a single binding site per circRNA. To increase biological relevance, we focused specifically on miRNAs involved in cancer-related pathways and validated their clinical significance through a KM plot and enrichment analysis to assess patient survival correlations. The following multi-step cross-validation approach thereby reduced the chance of false associations, mitigated the biases introduced by differences in datasets, and helped to overcome the limitations posed by small sample sizes.

While we focus on proposing that a circRNA-miRNA reciprocal expression combined with circRNA-miRNA sequence complementarity reflects a possible sponging mechanism, it is equally possible for other circRNAs and miRNAs, even with sequence complementarity, to exhibit no differential regulation or even parallel expression trends. This may be because circRNA sequesters and does not degrade miRNAs, thus reducing their availability to regulate downstream target genes, and might not directly alter miRNA expression levels. Our study emphasizes the importance of distinguishing between the total miRNA pool and the availability of free miRNAs capable of targeting downstream genes. In the case of tumor suppressor gene loss, circRNAs are expected to align in expression with their parental gene, leading to the proposed downregulation of circRNA expression. With reduced sponging, more miRNAs remain free to target downstream genes [156,157]. Here, the trend of circRNA-miRNA reciprocal expression proposed in these axes aligns with differential expression, as availability reflects expression levels. However, when an oncogene is activated and more circRNAs are generated, increased sponging may not consistently result in lower miRNA expression but rather a reduced availability of free miRNAs. It is also crucial to emphasize the complexity of molecular circRNA-miRNA regulatory networks. For instance, a study showed that CDR1as, known as ciRS-7, contains over 70 conserved binding sites for miR-7. Interestingly, ciRS-7 can be cleaved upon miR-671 binding in an AGO2-dependent manner, releasing the bound miR-7 [158,159]. This model illustrates a reciprocal expression relationship between CDR1as and miR-7, even though this reciprocal expression is not due to a direct interaction between the circRNA and miRNA of interest, highlighting the complex regulatory dynamics within this system. Thus, we are not stating in this work that the proposed reciprocal expression caused by direct sponging is universal for every circRNA and miRNA, but we are specifically interested in identifying circRNAs and miRNAs with reciprocal expression profiles and strong sequence complementarity that are implicated in tumor-related pathways for further validation in patients’ samples as cancer biomarkers. By contextualizing these axes within a multi-layered approach of oncogenic and tumor-suppressive pathways, we propose that the combined influence of circRNA on miRNA and miRNA on downstream gene expression offers a more robust biomarker potential than either component alone. Further, these proposed axes will be studied in vitro to gain insights into the underlying molecular mechanisms.

The ciRS-7/miR-7 interaction described above sheds light on another layer of system complexity and possible limitations. As previously discussed, circRNAs are generally highly stable molecules due to their covalently closed loop structure, which protects them from common degradation mechanisms such as deadenylation, decapping, and miRNA-induced degradation seen in mRNA targets [160,161]. While circRNAs seem largely immune to post-transcriptional degradation by miRNAs, some evidence indicates that this assumption may not universally apply and is highly context-dependent. In 2011, research demonstrated that miR-671 could facilitate the degradation of circRNA CDR1as in an AGO2-dependent manner [158]. More recently, miRNA-1224 was shown to cleave circRNA-Filip1l, also involving an AGO2-dependent mechanism [162]. Although evidence for miRNA-induced circRNA degradation remains limited, Appendix A lists various miRNAs that are proposed to interact with each of the five circRNAs in the examined axes. Several miRNAs, such as hsa-miR-145-5p, hsa-miR-203, hsa-miR-634, and hsa-miR-330-5p, are common across different circRNAs, suggesting a potential for widespread interactions that may influence similar cellular pathways and/or the expression levels of both the circRNA and miRNA of interest in each proposed axis. Interestingly, among miRNAs, we observe the presence of miR-671, which has previously been shown to degrade circRNA CDR1as through an AGO2-dependent mechanism. This miRNA may play a similar degradation role or perhaps a different regulatory function with hsa_circ_0088251 and hsa_circ_0001875. The shared miRNA-circRNA interaction network illustrates a complex regulatory system. However, it is important to note that the susceptibility to miRNA-induced degradation is likely specific to each circRNA, depending on its nature, function, and biological context. This selectivity underscores that while circRNA degradation by miRNAs is possible, it remains an exception rather than the rule, and the full mechanisms of circRNA degradation are not yet fully understood. Consequently, it would be premature to conclude that miRNA binding leads to circRNA degradation.

## 7. Conclusions

In this review, we highlight the role of circRNAs and miRNAs in BC, with a focus on their role in early detection and possible prevention.

Further, the critical impact of miRNA and circRNA expression on the initiation and progression of BC, both in vitro and in population studies, was highlighted. We proposed an epigenetic, big-data-based approach to identify mRNA/circRNA/miRNA axes that can be screened for early detection and potential prevention biomarkers of BC. Using two GEO datasets, we profiled differentially expressed circRNAs and miRNAs, identifying five mRNA/circRNA/miRNA axes. The established axes, of parent mRNA, derived circRNA, and target miRNA, propose that the reciprocal expression patterns of circRNAs and miRNAs could boost screening confidence compared to assessing the expression of each independently, enhancing what we refer to as “vertical confidence”. Additionally, including a higher number of initial genes in the screening process enhances “horizontal confidence” (Figure 2).

Screening for the expression of circRNA in body fluids, such as tissue-specific exosomes in blood, could serve as an early prediction tool for mRNA/protein expression variations in diseased tissues, which could otherwise be detectable only through tissue biopsies. Additionally, miRNA expression in body fluids or exosomes, when coupled with circRNAs, reflects the downstream tumor pathways involved in tumorigenesis with higher confidence. The use of mRNA(protein)/circRNA/miRNA axes in clinical applications could enhance non-invasive early cancer detection. The use of circRNAs and miRNAs could also facilitate personalized treatment. CircRNA/miRNA expression profiles could provide insights into the molecular profile of tumors. The proposed methodology is not only applicable in BC but can also be applied to other cancer types. Assessing the expression of early pre-tumorigenic mRNA(protein)/circRNA/miRNA axes provides risk assessment and early detection tools, increasing screening confidence and facilitating better implementation of preventive measures.

## Figures and Tables

**Figure 1 ijms-25-13056-f001:**
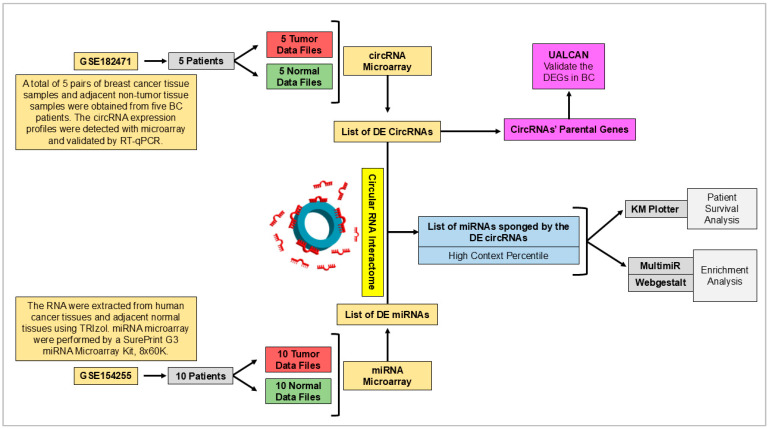
The methodological workflow used to identify the regulatory axes between circRNAs, miRNAs, and mRNAs in BC. To construct the circRNA/miRNA axes, we initially utilized the GEO dataset (accession number: GSE182471), which includes samples from 5 BC patients and 5 normal breast tissues. This dataset enabled the identification of differentially expressed circRNAs and the genes from which they originate. The expression of these genes in cancerous tissues was further validated through the UALCAN. We focused on the top differentially expressed circRNAs and identified their miRNAs proposed to be sponged by the selected circRNAs using the CircularRNA Interactome. Only miRNAs with a high context percentile (indicative of strong predictive binding potential) were selected. The involvement of miRNAs, proposed to be sponged by selected circRNAs, in breast tumorigenesis was tested through two in silico approaches. First, we analyzed miRNA expression via a GEO microarray dataset (accession number: GSE154255), which compared miRNA levels in 10 BC tissues against adjacent normal tissues (Table 3). Secondly, the survival implications of these miRNA expression levels were assessed using the KM Plotter. Lower survival rates in BC patients were correlated with selected miRNA expression levels. Further analysis using the “multimiR” package in R-studio helped identify the target genes of these miRNAs, while Webgestalt facilitated the enrichment analysis of miRNAs in cancer-related pathways.

**Figure 2 ijms-25-13056-f002:**
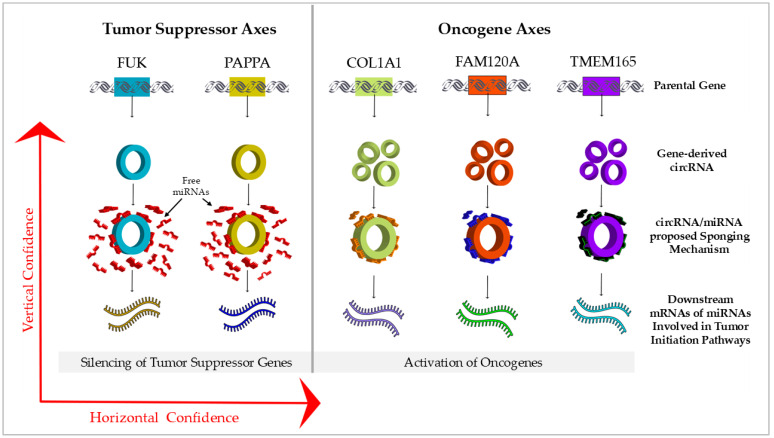
Vertical and horizontal confidence of oncogene and tumor suppressor axes. The integration of five mRNA/circRNA/miRNA axes, three ‘onco-axes’ and two ‘tumor suppressor axes’, may serve as a crucial indicator for the prediction and early detection of cancerous events. The combination of reciprocally modulated miRNA and circRNA aims to enhance the sensitivity and reliability of the screening process. Horizontal confidence, indicated by the combination of different oncosuppressor and tumor suppressor genes; vertical confidence, derived from multi-dimensional screening.

**Table 1 ijms-25-13056-t001:** List of miRNAs mined from literature as involved in breast cancer (BC) progression.

	miR-ID	Characteristic	Studied in	Target Gene	Reference
In vitro studies	miR-1	Tumor Suppressor miRNA	MCF-7 MDA-MB-435 MDA-MB-468 MDA-MB-231	CDK4 TMSB4X CNN3 TWF1 CORO1C WASF2	[65]
	miR-125-b	Tumor Suppressor miRNA	T47D SKBR3	MMP11	[66]
	miR-100	Tumor Suppressor miRNA	MDA-MB-231	VEGF	[67]
	miR-126	Tumor Suppressor miRNA	MCF-7 MDA-MB-231	VEGF ADAM9	[68]
	miR-145	oncomiR	MCF-7 T47D ZR75-1 BT-474 MDA-MB-453 BT-549 SK-BR-3 MDA-MB-231 MDA-MB-436	ROCK1	[69]
	miR-21	oncomiR	Hs578T MDA-MB-231 SK-BR-3 MCF-7	LZTFL1	[70,71,72]
	miR-155	oncomiR	MCF-7	TSPAN5	[73]
	miR-10b	oncomiR	MDA-MB-231	-	[74]
Population studies	miR-105-5p miR-767-5p miR-122-5p miR-187-3p miR-937-3p	Downregulated Downregulated Downregulated Upregulated Upregulated	African American (AA) and European American (EA) populations		[75]
	miR-125a-5p	Tumor Suppressor	Iranian population BC Tissues	STAT3	[76]
	miR-31 miR-362-3p miR-663	Downregulated Downregulated Upregulated	Lebanese Population BC Tissues	-	[77]
	miR-183 miR-182 miR-21 miR-200c miR-425-5p	Upregulated Upregulated Upregulated Upregulated Upregulated	Algerian and North African population BC Tissues	-	[78]
	miR-21	Upregulated	Indonesian Population Plasma	-	[79]
	miR-1246 miR-21	Upregulated Upregulated	Plasma-Derived Exosomes	-	[80]
	miR-148a miR-30c	Downregulated Downregulated	Blood		[81]

The miRNAs obtained from cell line studies were characterized between tumor suppressor and oncomiRs; meanwhile, miRNAs identified in population studies were characterized according to their pattern of regulation. The specific model of study is cited, along with the target genes of each miRNA when available.

**Table 2 ijms-25-13056-t002:** List of circRNAs mined from literature as involved in BC progression.

	circRNA-ID	Characteristic	Studied in	Reference
In vitro studies	circRPAP2 (hsa_circ_0000091)	Tumor suppressor circRNA	MCF-7 MDA-MB-231 BT549 BC tissues	[109]
	circ-1073	Tumor suppressor circRNA	MDA-MB-231 MDA-MB-468 BT-549 MCF-7 T47D ZR-75-1 SK-BR-3 BC tissues	[110]
	circ-VRK1	Tumor suppressor circRNA	BT474 MDA-MB-453 MDA-MB-231 BC tissues	[111]
	circ-ITCH	Tumor suppressor circRNA	MDA-MB-231 BT-549	[112]
	circWSB1	OncocircRNA	MCF-7	[113]
	circ-Dnmt1	OncocircRNA	MCF-7 cells BC tissues	[114]
Population studies	hsa_circ_0001831	OncocircRNA	BC cells	[115]
	circ-1073 circ-VRK1 hsa_circ_0068033 circTADA2A-E6 circRNA_103809	Tumor suppressor circRNA	BC tissues	[116]
	hsa_circ_0005046 hsa_circ_0001791	OncocircRNA	BC tissues	[117]
	hsa_circ_0001785 hsa_circ_0108942 hsa_circ_0068033	OncocircRNA NA NA	Peripheral blood of BC patients	[118]
	circSEPT9	OncocircRNA	BC tissues	[119]

Each circRNA is reported as originally written in its respective reference study; nomenclature varies among studies for name, ID, and alias. The name includes information about the parental gene, whereas the alias and ID are codes with 7 and 6 numbers, respectively. Each circRNA is characterized as a tumor suppressor circRNA or oncocircRNA. The model of study is also cited for each.

**Table 3 ijms-25-13056-t003:** Summary of patients and tumor characteristics used to build the circRNA/miRNA axes.

			Tissue	Gender	Age
circRNA	Non-coding RNA profiling by array Accession number: GSE182471	adjacent normal tissue	Breast tissue	Female	56 year
			Breast tissue	Female	38 year
			Breast tissue	Female	67 year
			Breast tissue	Female	47 year
			Breast tissue	Female	54 year
		breast cancer tissue	Breast tumor	Female	56 year
			Breast tumor	Female	38 year
			Breast tumor	Female	67 year
			Breast tumor	Female	47 year
			Breast tumor	Female	54 year
miRNA	Non-coding RNA profiling by array Accession number: GSE154255	adjacent normal tissue	HER2 adjacent1 normal tissue	Female	NA
			HER2 adjacent2 normal tissue	Female	NA
			HER2 adjacent normal tissue3	Female	NA
			luminal A adjacent normal tissue1	Female	NA
			luminal A adjacent normal tissue2	Female	NA
			luminal B adjacent normal1	Female	NA
			luminal B adjacent normal2	Female	NA
			TNBC adjacent normal1	Female	NA
			TNBC adjacent normal2	Female	NA
			TNBC adjacent normal3	Female	
		breast cancer tissue	HER2 cancer tissue1	Female	NA
			HER2 cancer tissue2	Female	NA
			HER2 cancer tissue3	Female	NA
			luminal A cancer tissue1	Female	NA
			luminal A cancer tissue2	Female	NA
			luminal B cancer1	Female	NA
			luminal B cancer2	Female	NA
			TNBC cancer1	Female	NA
			TNBC cancer2	Female	NA
			TNBC cancer3	Female	NA

Detailed characteristics of each patient are summarized in this table, as derived from the GEO database, providing an overview of the data used to establish the circRNA/miRNA axes. NA: Not Available.

**Table 4 ijms-25-13056-t004:** Summary of in silico tools that can be used to study circRNA/miRNA networks in cancer.

In Silico Tool Name	Used to Obtain	Link
UALCAN	Differential expression of selected miRNA in cancer	https://ualcan.path.uab.edu/ (accessed on 16 June 2024)
CircInteractome	The circRNA sequence and the gene from which it arises RNA-binding proteins (RBPs) binding to a circRNA List of miRNAs sponged by the circRNA of interest CircRNA-divergent primers and siRNAs specific to a circRNA of interest	https://circinteractome.nia.nih.gov/ (accessed on 10 April 2024)
KM Plotter	Predictive tool for cancer patients’ survival	https://kmplot.com/analysis/ (accessed on 1 November 2024)
Multi-miR	miRNA biological complementary targets, known as downstream mRNAs	https://www.targetscan.org/ (accessed on 12 July 2024) https://mybiosoftware.com/ (accessed on 12 July 2024)
Webgestalt	The enrichment analysis of a miRNA of interest	https://www.webgestalt.org/ (accessed on 15 October 2024)

A detailed overview of various bioinformatics tools used to perform diverse analyses related to circRNAs and miRNAs is summarized in this table. UALCAN offers insights into the differential expression of selected miRNAs in cancer. CircInteractome allows users to retrieve circRNA sequences, identify RNA-binding proteins (RBPs) that interact with circRNAs, and list miRNAs proposed to be sponged by specific circRNAs based on their sequence complementarity. Additionally, it helps generate divergent primers and siRNAs for experimental validation. KM Plotter serves as a predictive tool for analyzing cancer patients’ survival based on miRNA expression data, while multi-miR enables the identification of miRNA biological targets, including downstream mRNAs. Webgestalt facilitates the performance of enrichment analysis of miRNAs. Links to each tool are provided for direct access, making this table a comprehensive resource for cancer researchers focusing on circRNA/miRNA regulatory networks.

**Table 5 ijms-25-13056-t005:** Five mRNA/circRNA/miRNA axes that are involved in BC.

	Initial Proteins—Identified from BC Data	UALCAN *p*-Value	CircRNAs Derived from the Initial Gene	Up/Down circRNA	*p*-Value in circRNA Data of BC Patients	Fold Change of circRNA	microRNA Name	Up/Down miRNA	Context Percentile	*p*-Value in miRNA Seq Data of Normal vs. BC	Fold Change of miRNA	miRNA KM Plot *p*-Value	
1	*FUK*	0.00024094	hsa_circ_0002599	↓	0.00008	−0.901	hsa-miR-326-3p	↑	99	0.0283	2.7	0.033	Tumor Suppressor Axis
2	*PAPPA*	0.0047841	hsa_circ_0088251	↓	0.0001	−0.758	hsa-miR-326-3p	↑	86	0.0283	2.7	0.033	Tumor Suppressor Axis
3	*COL1A1*	1.62437 × 10^−12^	hsa_circ_0044556	↑	0.00013	1.275	hsa-miR-145-5p	↓	72	0.0000006	−3.40	0.051	Onco-Axis
4	*FAM120A*	4.5878 × 10^−6^	hsa_circ_0001875	↑	0.000225	1.798	hsa-miR-145-5p	↓	73	0.0000006	−3.40	0.051	Onco-Axis
5	*TMEM165*	1.62437 × 10^−12^	hsa_circ_0001414	↑	0.000368	1.130	hsa-miR-99a	↓	87	6.5 × 10^−6^	−3.42	0.000011	Onco-Axis

CircRNA microarray data were analyzed for 5 BC patients’ tissue samples relative to adjacent normal tissue to identify significantly differentially regulated circRNAs. The parental gene of each circRNA was verified using UALCAN to have a similar pattern of regulation; meanwhile, the target miRNAs of each circRNA were identified using CircInteractome. miRNASeq data were analyzed for 10 BC patients’ tissue samples relative to adjacent tissue samples to identify significantly differentially regulated miRNAs. Survival analysis for the expression of each miRNA was performed using a KM plot. CircRNAs and their target miRNAs were aligned for reciprocal expression. (↑) indicates upregulation, and (↓) indicates downregulation.

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
