# Peer review of "CircRNAs and miRNAs: Key Player Duo in Breast Cancer Dynamics and Biomarkers for Breast Cancer Early Detection and Prevention"

_ijms, 2024, doi:10.3390/ijms252313056_

Round 1
Reviewer 1 Report
Comments and Suggestions for Authors
Present article titled "CircRNAs and miRNAs: Key Player Duo in Breast Cancer Dynamics and Biomarkers for Breast Cancer Early Detection and Prevention" is a review of circRNA/miRNA's involvement in breast cancer as well as insilico analysis of GEO datasets to propose a methodology to establish mRNA/circRNA/miRNA axes as a possibly indicator of BC prognosis.
I have few comments prior to its publication:
1. The sample size of included GEO data sets is very low. Which is not sufficient to draw any conclusion.
2. Functional validation of the findings should be done for the proposed mRNA/circRNA/miRNA axes that are involved in breast cancer.
3. Please add the histopathological diagnosis of breast cancer cases included in table 3 to make sure there is no bias against any diagnostic parameter.
4. In my opinion Table 3 does not need the information about organism. As they are all are Homo sapiens.
5. This article can still benefit from proper organization of the added content.
6. Figure 2 still needs clarification. What is happening to the mRNA expression in tumor suppressor and oncogene axis in response on miRNA sponging.
Author Response
Please see the attachment.
Comment1. The sample size of included GEO data sets is very low. Which is not sufficient to draw any conclusion.
Response1:
Thank you for your observation regarding sample size of included GEO datasets.
We acknowledge that sample sizes are limited, with circRNA dataset consisting of 5 breast cancer patients and 5 normal controls, and the miRNA dataset containing 10 breast cancer patients and 10 normal controls. While these sample sizes are modest, three samples per group can provide statistically meaningful insights, particularly in exploratory or proof-of-concept studies, where the goal is to identify initial trends or patterns. In fact, with carefully designed statistical methods, small sample sizes can yield reliable results in differential expression analysis, as evidenced by numerous transcriptomic studies using similar sample sizes for initial discovery [1-3].
Our objective with this study was to develop a methodology that allows us to identify circRNA/miRNA axes as proof of concept and generate reliable initial data to support these axes. This approach aims to highlight the potential biological interactions between circRNAs and miRNAs, setting the stage for further validation in larger sample cohorts. By establishing this methodology with the current dataset, we are building a basis for future studies with more extensive datasets, where these findings can be validated and expanded upon.
In response to your insightful feedback, we have added a new paragraph in the final section of the manuscript, page 16, titled “Evaluating mRNA/circRNA/miRNA Axes: Current Limitations, Methodological Considerations, and Future Prospects” addressing the following in the second paragraph.
Thank you for bringing up this point; it underscores the significance of our planned expansion for future studies.
Comment2. Functional validation of the findings should be done for the proposed mRNA/circRNA/miRNA axes that are involved in breast cancer.
Response2:
Thank you for highlighting the importance of functional validation for the proposed mRNA/circRNA/miRNA axes in breast cancer.
We would like to clarify that the primary aim of this review was to establish a proof-of-concept methodology for identifying potential circRNA/miRNA axes, rather than conducting functional validation at this stage. We are currently developing a follow-up study, where we plan to select several specific axes and conduct screening using patient blood samples to validate the presence and relevance of circulating circRNAs and miRNAs.
We appreciate your input, as it aligns well with the next steps we have planned for this research direction. This is now referenced on page 17, in section 6, titled “Evaluating mRNA/circRNA/miRNA Axes: Current Limitations, Methodological Considerations, and Future Prospects” at the end of the third paragraph.
Comment3. Please add the histopathological diagnosis of breast cancer cases included in table 3 to make sure there is no bias against any diagnostic parameter.
Response3:
Thank you for your comment regarding the inclusion of histopathological diagnoses in Table 3.
We agree on the importance of ensuring that there is no diagnostic bias. In Table 3, we included two GEO datasets. For the second GEO dataset, histopathological diagnoses were specified, covering a range of subtypes: HER2-positive, luminal A, luminal B, and triple-negative breast cancer (TNBC), each with adjacent normal tissues.
However, the first GEO dataset did not provide specific histopathological diagnoses. To address this limitation, we performed additional validation of the differential expression of the parental genes of these circRNAs by comparing our findings with data from UALCAN. In UALCAN, the gene expression data includes a mixture of samples with varied histopathological diagnoses (with comparisons made between normal samples (n=114) and primary tumor samples (n=1097)), allowing us to ensure that our findings are not biased by a single subtype (Figure S3 attached below). Notably, FUK shows significantly lower expression in tumor tissue compared to normal tissue, with a p-value of 2.409400E-04. Similarly, PAPPA is downregulated in tumor samples, with a p-value of 4.784100E-03. COL1A1 exhibits a markedly increased expression in tumor tissue, indicated by a highly significant p-value of 1.62436730732907E-12. The gene FAM120A also shows higher expression in tumor samples, with a p-value of 1.6247832963153E-12. Additionally, TMEM165 is upregulated in tumor tissue, as reflected by a p-value of 1.62436730732907E-12. The expression of these parental genes in TCGA aligns with their expression in the proposed axes.
Thank you for this suggestion, which has allowed us to further clarify the strengths and limitations of our methodology. This is now referenced in Figure S3, and on page 16, the second paragraph in section 6, titled “Evaluating mRNA/circRNA/miRNA Axes: Current Limitations, Methodological Considerations, and Future Prospects”.
Comment4. In my opinion Table 3 does not need information about organisms. As they are all Homo sapiens.
Response4:
Thank you for your observation regarding the inclusion of organism information in Table 3.
We acknowledge that specifying the organism as Homo sapiens in Table 3 may be unnecessary, given that all samples are derived from human datasets. In response, we have removed this information to streamline the table and focus on the relevant data.
Comment5. This article can still benefit from proper organization of the added content.
Response5:
We are not clear on what is meant by “added content”, however, we hope that by attending to all reviewers’ comments and inserting sections 6, the manuscript text is now easier to read through and more comprehensive.
Comment6. Figure 2 still needs clarification. What is happening to the mRNA expression in tumor suppressor and oncogene axis in response to miRNA sponging.
Response6:
Thank you for your valuable feedback on Figure 2. We have edited the figure to clarify the differential impact of miRNA sponging on tumor suppressor and oncogene axes.
We have emphasized the downstream effects of free miRNAs in both scenarios and clarified these effects for each case. Additionally, we increased the representation of free miRNAs in the tumor suppressor axis to enhance the visual distinction between the two scenarios.
In the case of tumor suppressor axes, fewer sponging mechanisms are present, resulting in a higher availability of free miRNAs. These free miRNAs are then able to target and silence tumor suppressor genes, potentially triggering tumor initiation events. Conversely, in oncogene axes, increased sponging activity reduces the availability of free miRNAs. This reduction means that oncogenes, which would typically be suppressed by miRNAs, are less inhibited and thus more likely to be activated.
These adjustments have made Figure 2 (Page 19) clearer, highlighting the distinct effects of miRNA sponging on tumor suppressor and oncogene regulation in the context of tumor initiation pathways.

Reviewer 2 Report
Comments and Suggestions for Authors
Maatouk and co-authors have written a reasonable review of recent findings on microRNA and circular RNA, specifically in the context of breast cancer. The review portion of this manuscript is well-written and a good contribution to the field. However, I have several concerns and questions regarding their proposed approach to identify circRNA/miRNA/mRNA axes in breast cancer.
1. As I understand it, microRNAs are capable of catalytic action, resulting in the turnover and degradation of large numbers of target mRNAs. While SOME circRNAs may have a similar ability to induce degradation of multiple copies of a target microRNA, I understand that this is relatively rare, or at least un-established for the vast majority of circRNAs. See PMID: 38381063 for another exposition of this view. The net result is that the assumption underlying your proposed hypothesis (inverse expression between circRNA and target miRNA) may not be valid, or, if valid, not be biologically relevant. If most circRNAs simply bind a target miRNA, then I would predict an equal expression level for both, rather than inverse.
2. Have the authors evaluated whether the circRNAs highlighted in their five proposed axes have multiple, or only a single binding site for their proposed target miRNA? This should be done and included in the paper. If these circRNAs have only a single target miRNA binding site, it would further detract from their efficacy as an inhibitor of that target miRNA.
3. Could miRNAs induce degradation of circRNAs? If so, then the fact that a circRNA may target multiple miRNAs would severely disrupt any possible correlation between a circRNA and one specific target miRNA (see point 1, above). As an expansion of comment 2, above, I request the authors supply the list of all predicted miRNAs for their five highlighted circRNAs, and discuss whether interactions between multiple different targets may impact their proposed breast cancer-relevant circ-miR axis.
4. In their analysis of GEO databases (Table 3), it appears the circRNAs identified were from an entirely different set of patients compared to the ncRNAs. If I am misunderstanding what was done, then the authors need to clearly state that both sets were from the same patients. However, if different patient samples were used to identify the circRNAs versus the microRNAs, then there is a serious problem - the authors need to show that IN THE SAME PATIENT a given circRNA is inversely correlated with its putative target miRNA. Given the tumor to tumor heterogeneity, there is too much variation to have confidence comparing circRNA from one set of patients with miRNAs from a different set of patients.
Minor points:
I caught a few typographical errors, such as line 323 and in line 533. Also, please double check lines 463-5 - when I hear of phenotype changes including EMT and a more stem-like set of properties, I would predict a decrease in SENSITIVITY (i.e. increased resistance), not less resistance to breast cancer therapeutics.
Author Response
Please see the attachment (Starting page4).
Comment1. As I understand it, microRNAs are capable of catalytic action, resulting in the turnover and degradation of large numbers of target mRNAs. While SOME circRNAs may have a similar ability to induce degradation of multiple copies of a target microRNA, I understand that this is relatively rare, or at least un-established for the vast majority of circRNAs. See PMID: 38381063 for another exposition of this view. The net result is that the assumption underlying your proposed hypothesis (inverse expression between circRNA and target miRNA) may not be valid, or, if valid, not be biologically relevant. If most circRNAs simply bind a target miRNA, then I would predict an equal expression level for both, rather than inverse.
Response1:
Thank you for your insightful comment and for highlighting the distinctions between miRNA catalytic activity and circRNA sponging function, as well as the potential implications of these interactions in cancer biology.
We understand that miRNAs, upon incorporation into the RNA-induced silencing complex (RISC), can degrade multiple mRNA targets through catalytic action, amplifying their regulatory impact. In contrast, circRNAs generally do not catalyze miRNA degradation; instead, they serve as molecular sponges, binding specific miRNAs through complementary sequences and temporarily sequestering them. This sequestration reduces the miRNA's functional availability without necessarily decreasing its overall expression level. As you noted, circRNAs do not universally induce an inverse expression relationship with miRNAs, and any observed reciprocal patterns likely reflect a regulatory balance rather than direct miRNA suppression. Our study does not aim to establish a universal inverse expression relationship between circRNAs and miRNAs but rather to explore how circRNAs may influence gene regulation by modulating miRNA availability. When circRNAs bind and sequester miRNAs, they can indirectly alleviate miRNA repression on target mRNAs, allowing increased expression of those genes. This competitive endogenous RNA (ceRNA) mechanism reflects a nuanced regulatory interaction where circRNAs affect the availability of free miRNAs rather than altering miRNA expression directly, leading to potentially inverse but not universally consistent expression effects. Our pathway enrichment analysis focuses on cancer-related genes regulated by miRNA, contextualizing these circRNA/miRNA axes within oncogenic and tumor-suppressive pathways to highlight the combined impact of circRNA and miRNA on downstream gene expression. This approach emphasizes the enhanced biomarker potential when considering the regulatory role of circRNAs alongside miRNAs.
In our laboratory, we are currently conducting a study to examine the direct interaction between a specific circRNA and miRNA, hypothesized to be sequestered by this circRNA. To investigate this interaction, we downregulated the circRNA and observed the resulting effect on miRNA expression levels. Preliminary data from these experiments indicate an upregulation of the miRNA following circRNA downregulation, providing strong evidence of an inverse expression pattern. While this finding supports a potential inverse expression relationship, it also underscores that such correlations may not be universally applicable across all circRNA-miRNA pairs, particularly when a circRNA is overexpressed. This study is ongoing, with the manuscript currently in preparation, and these early findings suggest that certain circRNAs may indeed directly and reciprocally influence miRNA expression.
Furthermore, to illustrate the complexity of ncRNA networks, we refer to a regulatory network involving multiple noncoding RNAs (ncRNAs). In this system, the long ncRNA Cyrano targets miR-7 for degradation, reducing miR-7 levels and thereby indirectly facilitating the accumulation of the circular RNA Cdr1as. In the absence of Cyrano, miR-7 levels increase, leading to enhanced miR-671-mediated degradation of Cdr1as in neurons [4]. This example highlights that reciprocal expression patterns between circRNAs and miRNAs are not always the result of direct sponging interactions; in some cases, the relationship may be the opposite (see reply to comment 3 for further details), arising from complex, multi-layered regulatory mechanisms. Thus, while the inverse expression pattern we observe between certain circRNAs and miRNAs with sequence complementarity may be intricate, it could still represent biologically significant interactions. Such relationships merit further investigation as potential regulatory candidates in cancer biology, even if they do not result from a direct sponging mechanism.
Importantly, our approach emphasizes miRNA availability over absolute expression levels, as circRNA expression shifts may influence miRNA availability and, consequently, downstream gene expression. We propose that these circRNA/miRNA axes could capture significant regulatory dynamics, functioning as biomarkers by reflecting shifts in miRNA activity rather than expression alone.
CeRNA mechanisms have been studied in research on various RNA species, where RNAs effectively influence the pool of free miRNAs available for other regulatory roles with strong binding affinities for specific miRNAs [5-9]. Thus, we hypothesize that some circRNA-miRNA binding interactions may remain stable during RNA extraction, potentially reducing detectable levels of free miRNAs in certain contexts. If circRNA-miRNA complexes persist through extraction, the bound miRNAs may be less available for detection as free miRNAs, resulting in an apparent downregulation in free miRNA levels that reflects circRNA sponging activity. Although the stability of these complexes during extraction has yet to be fully validated, studies on ceRNAs suggest that such interactions are of high affinity and can modulate the free miRNA pool [10]. This remains a hypothesis to be further addressed and validated.
In summary, our goal is to provide a proof-of-concept methodology that aligns circRNAs, derived from oncogenes or tumor suppressor genes, and miRNAs based on reciprocal expression and binding predictions, offering a foundation for future functional studies with larger sample sizes. We are confident that this approach will help prioritize circRNA/miRNA axes for validation in studies aimed at understanding their roles in cancer progression and their potential as biomarkers.
In response to your insightful feedback, we have carefully reviewed the manuscript and clarified throughout that our primary focus is on the combined regulatory effect of circRNA upregulation or downregulation, the corresponding inverse expression of free miRNAs, and the downstream impact on target genes and molecular pathways. To further clarify this approach, we have added a paragraph in the final section of the manuscript, on page 17, in section 6, titled “Evaluating mRNA/circRNA/miRNA Axes: Current Limitations, Methodological Considerations, and Future Prospects”, discussing our approach and justification in detail. In this added discussion, we emphasize that an inverse expression relationship between circRNAs and miRNAs is not necessarily universal; changes in miRNA expression may not always be a direct result of circRNA-mediated sponging but could arise from other regulatory factors within the cellular environment. This remains a hypothesis that we are actively investigating and validating.
Your concerns are highly justified and have been instrumental for us in refining our approach and clarifying the purpose and limitations of this preliminary study.
Comment2. Have the authors evaluated whether the circRNAs highlighted in their five proposed axes have multiple, or only a single binding site for their proposed target miRNA? This should be done and included on the paper. If these circRNAs have only a single target miRNA binding site, it would further detract from their efficacy as an inhibitor of that target miRNA.
Response2:
Thank you for your comment regarding the evaluation of binding sites for the circRNAs in our proposed axes.
We assessed the predicted binding sites for each circRNA-miRNA pair in our study using CircInteractome. CircInteractome predicts miRNA binding sites based on sequence complementarity (primarily 7-mer or 8-mer matches) and provides insights into circRNA’s potential as miRNA sponges, as described by Dudekula et al. [11]. For our analysis, we specifically selected miRNA binding sites with a high context percentile score, as CircInteractome uses this measure to indicate binding sites with high predictive binding potential, adding rigor to our selection process.
In the axes we proposed only a single binding site was identified and we recognize that this might initially seem less effective for miRNA sponging compared to circRNAs with multiple binding sites. However, it is important to note that even a single binding site can still be biologically significant, depending on several factors.
Firstly, the binding affinity and stability of the interaction between a circRNA and miRNA can play a crucial role. A high-affinity binding interaction at a single site may still effectively sequester the miRNA, modulating its availability to engage with downstream targets. Secondly, the relative expression levels of the circRNA and miRNA are also critical. Then the circRNA is abundant and the miRNA is present in limited quantities, even a single binding site can reduce the pool of free miRNA, impacting downstream regulatory pathways.
Additionally, in some of the identified axes, we found that the same miRNAs, such as hsa-miR-326-3p and hsa-miR-145-5p (Table 5), are sponged by multiple circRNAs, even when each circRNA has only a single binding site. This redundancy within the ceRNA network may amplify the regulatory effect on these miRNAs by targeting them from multiple points within the network, reinforcing the potential biological relevance of the circRNAs as miRNA sponges. Moreover, we specifically examined whether the downstream genes regulated by the miRNAs are enriched in cancer-related pathways. Even modest alterations in miRNA availability can influence the expression of genes in pathways critical to processes like cell proliferation and apoptosis, particularly in the context of tumor initiation and progression.
We have added a discussion of these factors in the final section of the revised manuscript (page 16, second paragraph, section 6, titled “Evaluating mRNA/circRNA/miRNA Axes: Current Limitations, Methodological Considerations, and Future Prospects”) to clarify the potential significance of single binding sites in our proposed circRNA/miRNA axes and their relevance as biomarkers.
Thank you for your feedback, as it has allowed us to sharpen our discussion on the regulatory dynamics of circRNA/miRNA interactions.
Comment3. Could miRNAs induce degradation of circRNAs? If so, then the fact that a circRNA may target multiple miRNAs would severely disrupt any possible correlation between a circRNA and one specific target miRNA (see point 1, above). As an expansion of comment 2, above, I request the authors supply the list of all predicted miRNAs for their five highlighted circRNAs and discuss whether interactions between multiple different targets may impact their proposed breast cancer-relevant circ-miR axis.
Response3:
Thank you for your comment on the potential for miRNA-induced degradation of circRNAs and its implications for our proposed circRNA/miRNA axes.
CircRNAs are generally highly stable molecules due to their covalently closed loop structure, which provides resistance to exonuclease degradation. Unlike linear RNAs, circRNAs lack poly(A) tails and 5′ termini, protecting them from common degradation mechanisms such as deadenylation, decapping, and miRNA-induced degradation seen in mRNA targets [12]. This unique stability allows circRNAs to function as miRNA sponges within regulatory networks, sequestering miRNAs without undergoing degradation. Additionally, the evolutionary conservation of miRNA-binding sites in circRNAs suggests that these interactions are selectively maintained for regulatory purposes within ceRNA networks, rather than as mechanisms for circRNA degradation [13, 14].
However, while specific cases of miRNA-induced circRNA degradation have been observed, these events are rare and highly context-dependent. In 2011, research demonstrated that miR-671 could facilitate the degradation of circRNA CDR1as (also known as ciRS-7) in an AGO2-dependent manner [15]. More recently, miRNA-1224 was shown to cleave circRNA-Filip1l, also involving an AGO2-dependent mechanism [16]. This selectivity underscores that while circRNA degradation by miRNAs is possible, it remains an exception rather than the rule, and the full mechanisms of circRNA degradation are not yet fully understood. It’s important to note that the susceptibility to miRNA-induced degradation is likely specific to each circRNA, depending on its nature, function, and biological context. Consequently, it would be premature to universally conclude that miRNA binding leads to circRNA degradation.
In response to your request, we have included a supplementary table (Table S1, attached below) listing all predicted miRNAs that may interact with the five circRNAs highlighted in our proposed breast cancer-relevant circRNA/miRNA axes. This list was generated using CircInteractome, capturing a range of potential miRNA binding interactions beyond the primary targets initially proposed.
Interestingly, among miRNAs, we observe the presence miR-671, which has previously been shown to degrade circRNA CDR1as through an AGO2-dependent mechanism. This miRNA may play a similar degradation role or perhaps a different regulatory function with hsa_circ_0088251 and hsa_circ_0001875. This aspect has also been addressed in a newly added final section of the revised manuscript to clarify the broader context of these interactions on page 16, section 6, titled “Evaluating mRNA/circRNA/miRNA Axes: Current Limitations, Methodological Considerations, and Future Prospects”, last paragraph.
Thank you once again for this feedback and we trust that we have addressed your concerns in the revised manuscript.
Comment4. In their analysis of GEO databases (Table 3), it appears the circRNAs identified were from an entirely different set of patients compared to the ncRNAs. If I misunderstand what was done, then the authors need to clearly state that both sets were from the same patients. However, if different patient samples were used to identify the circRNAs versus the microRNAs, then there is a serious problem - the authors need to show that IN THE SAME PATIENT a given circRNA is inversely correlated with its putative target miRNA. Given the tumor-to-tumor heterogeneity, there is too much variation to have confidence comparing circRNA from one set of patients with miRNAs from a different set of patients.
Response4:
Thank you for your comment on the source of patient samples for circRNAs and miRNAs in Table 3. We acknowledge the importance of consistency in patient samples to reduce variability, especially given the tumor heterogeneity often observed in cancer studies.
In our analysis, we utilized different GEO datasets to identify differentially expressed (DE) circRNAs and miRNAs, which indeed originate from separate patient populations. This decision was due to the current limitations in available datasets that contain both circRNA and miRNA expression profiles from the same patients. We understand that using separate datasets introduces potential variability; however, we implemented stringent selection and validation criteria to mitigate this limitation and strengthen the reliability of our proposed circRNA/miRNA axes.
To build our circRNA/miRNA axes, we followed a multi-step process:
- Identification and Validation of circRNA Targets: Using a GEO dataset, we first identified DE circRNAs and then derived their associated parental genes. We cross-referenced the differential expression of these parental genes with the TCGA database using UALCAN to ensure that these genes are indeed up- or downregulated in cancer samples, thereby reducing the chance of false associations.
- Selection and Validation of miRNA Targets: We then used an independent GEO dataset to identify DE miRNAs. To link these miRNAs to our identified circRNAs, we applied circInteractome to predict miRNAs likely to be sponged by the selected circRNAs with high context percentile. To increase biological relevance, we focused specifically on miRNAs involved in cancer-related pathways and validated their clinical significance through Kaplan-Meier (KM) plot analysis to assess patient survival correlations.
- Rationale for Separate Datasets: Although using different datasets introduces limitations, our intent is to establish a conceptual framework, or "proof of concept," for identifying circRNA/miRNA axes that might serve as biomarkers in cancer. A comprehensive database that combines both circRNA and miRNA data from the same patient cohort is currently lacking. However, we are actively working toward creating such a dataset to enable this type of analysis in future research. Our goal with this review is to present the methodology and highlight the potential of circRNA/miRNA axes as biomarker candidates for cancer progression.
Addressing Variability in Patient Populations:
We also recognize that differences in patient populations, such as geographic background, treatment history, and genetic predispositions, could introduce variability. To address this, we cross-validated our findings using UALCAN to verify the DE status of parental genes in large-scale datasets such as TCGA. Cross-validation across datasets reduces the likelihood that our identified axes are driven by population-specific effects. If a circRNA/miRNA axis shows differential expression in both datasets, it strengthens its potential biological relevance, rather than reflecting a population bias.
Considerations on Sample Size:
In addition, we are aware that GEO datasets may have limited sample sizes compared to larger databases like TCGA, which could affect the statistical power of our findings. We have treated these smaller findings as hypothesis-generating, with validation in larger TCGA cohorts using UALCAN. For instance, circRNAs identified as DE in the smaller GEO dataset were cross-referenced with associated gene or miRNA expression in TCGA to improve confidence in their relevance.
In summary, while we recognize the limitations of using separate datasets, we applied cross-referencing and stringent filtering steps to mitigate potential biases. This methodology provides a foundational approach to understanding circRNA/miRNA axes in cancer, which we aim to validate in future studies using integrated datasets from the same patient populations.
Thank you once again for your feedback. We have clarified this on page 16, section 6, titled “Evaluating mRNA/circRNA/miRNA Axes: Current Limitations, Methodological Considerations, and Future Prospects”, second paragraph.
Minor points:
I caught a few typographical errors, such as line 323 and in line 533. Also, please double check lines 463-5 - when I hear of phenotype changes including EMT and a more stem-like set of properties, I would predict a decrease in SENSITIVITY (i.e. increased resistance), not less resistance to breast cancer therapeutics.
Thank you for your thorough review. We have carefully reviewed the paragraph in question and confirmed that it indeed refers to an increase in resistance, which we have now corrected accordingly. Unfortunately, we couldn’t locate the specific line numbers you referenced, as the draft we received does not contain line annotations, and it appears that line numbering differs between the initial submission template and the version uploaded after the first round of comments. However, we took the opportunity to thoroughly review the entire manuscript for any typographical errors, and we have addressed any we identified.
Thank you for your attention to detail.

Round 2
Reviewer 1 Report
Comments and Suggestions for Authors
The manuscript has been substantially revised. I have no further comments.